# Voisin Rational Grazing as a Sustainable Alternative for Livestock Production

**DOI:** 10.3390/ani11123494

**Published:** 2021-12-08

**Authors:** Luiz C. Pinheiro Machado Filho, Hizumi L. S. Seó, Ruan R. Daros, Daniel Enriquez-Hidalgo, Adenor V. Wendling, Luiz C. Pinheiro Machado

**Affiliations:** 1LETA, Laboratory of Applied Ethology, Department of Zootechny and Rural Development, Federal University of Santa Catarina, Florianópolis 88034-001, Brazil; hizumi@hotmail.com (H.L.S.S.); pinheiro@cca.ufsc.br (L.C.P.M.); 2Graduate Program in Animal Science, School of Life Science, Pontifícia Universidade Católica do Paraná, Curitiba 80215-901, Brazil; r.daros@pucpr.br; 3Bristol Veterinary School, University of Bristol, North Somerset, Langford BS40 5DU, Somerset, UK; daniel.enriquez@bristol.ac.uk; 4Sustainable Agriculture Sciences, Rothamsted Research, North Wyke, Okehampton EX20 2SB, Devon, UK; 5Federal Institute of Paraná, Campus Palmas, Palmas 85555-000, Brazil; adenor.wendling@ifpr.edu.br

**Keywords:** multi-paddock management, regenerative management, agroecological pasture management, multispecies pasture, biodiversity

## Abstract

**Simple Summary:**

To meet current global food supply challenges, researchers and technicians are searching for alternative livestock systems that promote highly sustainable animal productivity and farm profitability, while having a positive environmental footprint. In this narrative review, we highlight one such system, Voisin Rational Grazing (VRG). VRG is a regenerative livestock farming system proposed by French scientist Andre Voisin in the 1950s and further developed in Brazil in the 1970s. VRG has been applied in many countries with vastly different ecosystems. Like other regenerative systems, VRG provides a range of ecosystem services, including negative net carbon emission, reduced soil erosion, and increased biodiversity. Because VRG is also focused on animal performance, farmers applying VRG are more resilient against the adversities confronting farmers practicing more conventional farming systems. VRG requires a paradigm shift from the farmers and thus its uptake may be hindered if there is not enough support within the community. Here, we provide a comprehensive overview of VRG, along with its benefits and constraints.

**Abstract:**

Current livestock practices do not meet current real-world social and environmental requirements, pushing farmers away from rural areas and only sustaining high productivity through the overuse of fossil fuels, causing numerous environmental side effects. In this narrative review, we explore how the Voisin Rational Grazing (VRG) system responds to this problem. VRG is an agroecological system based on four principles that maximise pasture growth and ruminant intake, while, at the same time, maintaining system sustainability. It applies a wide range of regenerative agricultural practices, such as the use of multispecies swards combined with agroforestry. Planning allows grazing to take place when pastures reach their optimal resting period, thus promoting vigorous pasture regrowth. Moreover, paddocks are designed in a way that allow animals to have free access to water and shade, improving overall animal welfare. In combination, these practices result in increased soil C uptake and soil health, boost water retention, and protect water quality. VRG may be used to provide ecosystem services that mitigate some of the current global challenges and create opportunities for farmers to apply greener practices and become more resilient. It can be said that VRG practitioners are part of the initiatives that are rethinking modern livestock agriculture. Its main challenges, however, arise from social constraints. More specifically, local incentives and initiatives that encourage farmers to take an interest in the ecological processes involved in livestock farming are still lacking. Little research has been conducted to validate the empirical evidence of VRG benefits on animal performance or to overcome VRG limitations.

## 1. Introduction

Countries around the world have faced limitations in using the planet’s biophysical resources to meet basic needs. Achieving a high standard of living for all would require the use of biophysical resources two to six times greater than the level considered sustainable [1], depleting our resources and threatening life on the planet. Agriculture plays a central role in providing food resources, but the current agricultural model has caused serious environmental and social impacts [2].

According to the UNCCD (United Nations Convention to Combat Desertification; https://www.unccd.int/message-land-and-soil), the planet loses 28–75 billion tons of fertile soil to erosion annually. Livestock has been considered responsible for 14.5% of anthropogenic greenhouse gases (GHG) emitted, of which meat and dairy products represent 41% and 21%, respectively [3]. The current agricultural model, which is based on the “Green Revolution”, has also promoted massive rural exodus, changing the rural landscape from its cultural richness and biodiversity to the monotony of monocultures in large farms [4].

Promoting sustainable livestock production systems is imperative, and the current challenge is to combine intensification of animal production and maintenance of ecosystem services [5]. Well-managed pastureland provides a range of ecosystem services [6,7], but perhaps this is not enough. A key aspect of sustainable livestock production is to identify the best management practices to optimize environmental services, while supporting farmers’ profitability [8]. Maintaining high animal productivity is in line with farmers’ motivations [9,10] and may help bridge the gap between the trade-offs of confinement and extensive systems. Ecologically based systems require a paradigm that focuses on agroecological processes involving soil, plants, and animals in order to optimise the use of renewable resources [10,11]. This means shifting away from intensive agriculture heavily based on fossil fuel inputs and moving toward intensification based on pasture management and solar energy [12].

Permanent meadows and pastures encompass about 67% (33.6 million km^2^) of the world’s agricultural land [13]. This large area is probably sufficient to sustain not only wildlife, but also the current livestock population of ruminants: 1.5 billion head of cattle, 204 million head of buffaloes, 1.24 billion sheep, and 1.09 billion goats [13] in sustainable, regenerative systems. Many types of systems may be considered ecologically based or regenerative: holistic grazing management [14], adaptive multi-paddock grazing management (AMP; [15]), management intensive grazing (MIG; [16]) and Voisin Rational Grazing (VRG; [12,17]). All these systems rely on pastoral ecosystems and share the basis of agroecological reasoning, notwithstanding there are different reasons for applying one system instead of another. While some systems are applied mostly with the aim of land recovery, others focus on increased animal productivity.

Another agriculture is possible and necessary, with sustainability as a fundamental guide. For ruminants, in particular, Voisin Rational Grazing (VRG) [12,17] addresses sustainability in all its dimensions: economic, energetic, productive, social, cultural, environmental, and animal welfare, bolstering the pastoral ecosystem. VRG is part of an agricultural system based on ecology, compatible with the integration of livestock and agriculture, through the rotation of pasture and crop areas, the use of cover crops without disturbing the soil and without the use of pesticides. It optimises the use of endogenous resources and minimises external dependence, reducing costs and increasing profitability.

In this narrative review, we argue that VRG delivers high animal productivity, while improving other ecosystem services. VRG focuses on four basic principles (recovery period, occupation time, maximum performance, and regular performance principles; see Section 2) that account for forage growth and management, as well as animal requirements. Because it is based on simple ecological principles, VRG can be applied in a range of ecosystems, making VRG a universally viable option for the future of animal production worldwide. The objective of this review is to describe the key management practices of VRG and explore their potential for increasing productivity, improving ecosystem services and farm resilience, in addition to mitigating the negative effects of animal production. In the first section, we highlight the principles that guide VRG and the refinements proposed by several researchers to improve its productivity and identify other grazing systems that are similar to VRG. Secondly, we demonstrate how VRG responds to current global challenges in livestock management. Thirdly, we discuss the barriers against the adoption of VRG. We finish with future directions and limitations.

## 2. Voisin Rational Grazing and Its Four Principles

VRG can be defined as a rational method for managing the soil–plant–animal complex through direct grazing and well-planned pasture rotation. The system was first described in André Voisin’s book entitled *Productivité de l’herbe* (1957; [17]). Among other works, he later published *Dynamique des herbages* (1960; [18]) to complement the concept of pasture management. Voisin’s work was first implemented in South America in 1964 by the agronomist Nilo Ferreira Romero on his farm called Conquista located in Bagé, southern Brazil [12,19,20].

Further, the Brazilian agronomist and professor, Luiz Carlos Pinheiro Machado, coined the term “Pastoreio Racional Voisin” (Voisin Rational Grazing) and introduced advances in the system, such as the concept of bringing water to the animal and not the animal to the water, the need for shade in the paddocks and the concept of dividing the area with square paddocks and internal and external corridors designed to facilitate the flow of animals and prevent soil erosion. Pinheiro Machado left his book *Pastoreio Racional Voisin* (2004; [12]) as his main legacy. Developed in Brazil in the 1970s and 1980s, VRG spread to other countries in the Americas. The American professor Bill Murphy brought VRG to the USA from Brazil and began to teach and research the method at the University of Vermont, USA. In his work, professor Murphy used the term “MIG” (Management Intensive Grazing; [16]). More recently, in December 2020, the French Academy of Agriculture acknowledged and recognized the work of Andre Marcel Voisin in a Webinar promoted by the *Association pour l’étude de l’histoire de l’agriculture* (https://www.academie-agriculture.fr/actualites/academie/seance/academie/seance-organisee-par-laeha-andre-voisin-controverses-autour-de; 3 August 2021).

The VRG system follows four “laws” (principles) of rational grazing, as first enunciated by Voisin (1957; [17]) and summarized below:

**(I) “First Law”—recovery period principle:** Before a sward, sheared with the animal’s teeth, can achieve its maximum productivity, a sufficient interval must have elapsed between two successive shearings to allow the grass (1) to accumulate in its roots the reserves necessary for a vigorous spurt of regrowth and (2) to produce its “blaze of growth” (or highest daily yield per hectare).

Time of recovery period is always variable and should provide an optimum post-grazing period that enables full plant recovery after the following grazing bout. This optimum recovery period (ORP; Figure 1) can be defined as the moment when the acceleration of pasture growth curve is equal to zero; the moment of maximum herbage growth rate [21], which has been related to the regrowth moment when light interception reaches 95% [19]. The ORP coincides with the maximum accumulation rate of protein, energy, and organic matter digestibility in herbage [20,22,23]. On the other hand, after the plant has reached its ORP, it rapidly redirects nutrients and energy to enter into the reproductive stages, followed by the decline of herbage mass growth rate, drop in leaf to stem ratio, and severe reduction of herbage quality [23,24,25]. The ORP can be determined by the plant’s phenological stages, right before it directs its energies towards flowering. For instance, at the paddock level, ORP occurs when (i) plants begin stem elongation, (ii) the flag leaf emerges [26], (iii) boot stages occur, which is common in most grass species, and/or (iv) the first emerged leaves become senescent or when 30 to 50% of plants are in flowering stage for temperate legumes [12].

For the first principle to be achieved, VRG does not follow a pre-established sequential pasture rotation scheme. Instead, it uses each paddock at the moment its ORP has been reached by the targeted pasture species within the paddock or field to be grazed. Species growth is dependent on soil characteristics and environmental conditions, particularly the lack of adequate soil moisture or precipitation [27]. Therefore, ORP for a given species is site-specific and varies along the grazing season [17]. In order to attain ORP for a given species at any moment of the season, it is necessary to determine a set number of paddocks according to the longest ORP of the intended species. The number of paddocks is key to allow for a high degree of control over the timing of occupancy of a grazed area [12,15]. Most commonly, a paddock is grazed using a mean ORP when most of the desired species have attained their ORP. However, the ORP of a particular species can be targeted to allow it to increase its presence in the paddock [12]. Moreover, targeting the ORP of the most productive and/or best nutritive value plant species may promote ideal conditions to maximise overall annual herbage production and nutritive value [11], in turn hampering the survival of undesirable species or weeds [26,27,28]. Such effects are related to the fact that when the plant is cut at its ORP, there is the best combination of accumulated reserves and the lowest fibre content in the plant tissue [28]. Thus, when cut at this point, it will have a faster and more vigorous regrowth than other plants that have not attained their ORP and will have fewer reserves to promote a vigorous regrowth. More mature plants that have passed their ORP will have already redirected some of the accumulated reserves to the flower and seed formation, and will be less palatable to animals due to a higher fibre content. As a consequence, the grazed plant will have a higher senescent residue, with greater respiration and lower photosynthetic rate during the regrowth [29], reducing their competitiveness compared with plants cut at their ORP. Pratensis plants, as Voisin (1957) called them, or plants growing in meadows that co-evolved with ruminants, have high tolerance to grazing. Compared with a 60-day cutting interval, frequent defoliation reduced root and shoot biomass in species with high tolerance to grazing, but not in species with low grazing tolerance [30]. Thus, it is expected that in a multispecies pasture cut at its ORP, this characteristic would favour the presence of high-tolerance grazing species and reduce the participation of non-grazing species.

**(II) “Second Law”—occupation principle:** The total occupation period on one paddock should be sufficiently short for a grass sheared on the first day (or at the beginning) of occupation not to be cut again by the teeth of these animals before they leave the paddock.

This principle is tightly related to the first law in order to prevent grazing of plant regrowth. In this sense and as a rule of thumb, the animals should not stay more than 3 days grazing the same paddock, ideally allocating one or even two paddocks per day [15,31]. However, this period is site-specific. For example, in tropical areas where herbage growth rates are high, the period to avoid grazing of herbage regrowth should be shorter. On the other hand, in conditions where herbage growth is slow or dormant, as in summer in Mediterranean climates or winter in temperate climates, these periods can be extended [32]. To obtain short occupation times, it is necessary to use high stocking densities, which results in concentrated manure deposits [10], promoting large flows of readily available organic matter to activate soil biocenosis [33] that will provide the required nutrients to ultimately guarantee a fast growth rate of herbage after defoliation and during the growing season [12].

**(III) “Third Law”—maximum performance principle:** The animals with the greatest nutritional requirements must be helped to harvest the greatest quantity of grass of the best possible quality.

To achieve maximum herd performance, animals of higher nutritional demand should be allowed the herbage of greatest nutritive value. Herbage nutritional value is greatest at the top fraction of the canopy and lowest at the lower fractions [31,32]. Thus, to follow this rationale one may separate groups of animals by their nutritional requirements. For example, lactating animals (higher nutritional demand) may enter a fresh paddock while non-lactating animals (lower nutritional demand) may enter that same paddock shortly after the lactating herd has left to a new fresh paddock. This management is deemed first and second grazing groups [34].

**(IV) “Fourth Law”—regular performance principle:** If a cow is to give regular milk yields she must not stay any longer than three days on the same paddock. Yields will be at their maximum if the cow stays on one paddock for only one day.

Animals should be offered herbage of consistent quality to maximise their performance and avoid unstable productivity. Although ruminant animals are resilient to irregular feed offering, it does decrease overall productivity and promote irregular performance [35]. Thus, in line with the second principle, to achieve the best and most consistent performance, lactating dairy cows, for example, should be moved to a fresh pasture after each milking (e.g., twice a day). Likewise, finishing steers should not stay more than one day grazing the same paddock. To ensure proper tight grazing, following the first grazing group, a category of animals with lower requirements should occupy the paddock for similar length of time [12]. Tight grazing regime promotes herbage regrowth of new photosynthetically active tissue with higher leaf to tiller rates. Thus, increased grazing severity maintains high-nutritive value grass [36].

The dynamic and complete observance of VRG principles is key to attaining maximum system production efficiency, including positive responses in the quality of food produced [17]. The application of the four principles must be dynamic, dialectic, and constantly evaluated, but without fixed rules, fixed times, or fixed stocking densities. However, this requires good planning. It is a dynamic management process of the soil–plant–animal complex with holistic evaluation throughout the pastoral ecosystems. The key aspect to achieve such management is time. ORP never has the same length, therefore the sequence of paddocks’ use is not repeated in consecutive grazing seasons. Likewise, occupation time varies with pasture productivity over the season. In the first–second group dynamic, the first group will leave to a fresh paddock when all pasture of the second group´s paddock is consumed. Therefore, the second group defines the moment of paddock change for both groups. These management principles (Table 1) are oriented toward satisfying both herbage and animal requirements [17].

Among different grassland management systems that have been described, we see close similarities between VRG and the adaptive multi-paddock system (AMP; [15]), as well as management-intensive grazing (MIG) or management-intensive rotational grazing (MIRG) [16]. These management methods follow principles very similar to the four previously described for VRG, thus we will use the research outputs from studies undertaken using these systems, along with studies done with VRG, to support the rationale and mechanism underlying VRG responses.

## 3. Voisin Rational Grazing Refinements and Implementation

Maximum efficiency of pasture and animal performance is obtained with the dynamic and complete application of the four principles of rational grazing [17], which, together, presuppose the division of an area into multiple paddocks. This is achieved by planning the use of pastures by calculating the number of paddocks, estimating the evolution of the animal stocking rate and the flow of animals within the farm, and allocating shade, drinking water, and forage species. The concept of square paddocks, along with the perimeter corridors and the use of water troughs within paddocks, were important advances in the application of VRG [12]. Modern VRG systems are designed with multiple corridors, allowing more than one choice to move from one paddock to another, and with water troughs available in all paddocks. This design avoids excessive use of the same corridor for herd movement, which may cause soil erosion, and ensures water supply for all animals. An example of such design can be seen in Figure 2a. When no fixed rotational pasture schedule is used, the grazing area assumes a non-uniform pattern, sometimes called a “chess” pattern (Figure 2b; [12]).

Overall, the integration of the four principles results in an intensive grazing system through the use of first–second grazing groups that graze for short periods of time in a small area, resulting in a high stocking density. This approach, when combined with targeted ORP, maximises both herbage production and nutritive value, and ultimately maximises animal productivity along a series of additional ecosystem services (see below).

Under VRG, the animals are less selective in their grazing behaviour, becoming more voracious. This means that animals graze almost all species available, leaving few unconsumed species in the sward [11,14,25,27]. As plants are repeatedly cut in this way, there is a tendency to reduce the presence of non-grazing species, although not decreasing diversity. Azevedo et al. [37] reported 81 plant species from 23 different families per square meter in a VRG in Bom Retiro, SC, Brazil. If the high animal density leads to a more voracious ingestive behaviour, applying excessive stocking density may affect overall animal behaviour and welfare. For example, comparing two high stocking densities (200 cows/ha and 500 cows/ha), cows in the lower stocking density group performed more grazing and had less aggressive behaviour than cows in the higher stocking density [38]. Furthermore, the use of a very high stocking density may require three or more paddock changes per day, increasing labour.

Paddocks must be designed with water troughs and shade access. In VRG systems for dairy and beef cattle, readily available water troughs increase animal productivity [39,40]. Like any social species, ruminants form social hierarchies that influence group behaviour; subordinate animals may not have fulfilled their physiological needs before the dominant, and high-ranking animals start moving away from the water resources, for example. Water, shade, and other valuable resources (e.g., mineral supplements) located within the paddock must be planned to minimise the herd dominance effect. As a demonstration of this phenomenon, researchers have compared the effect of water trough location on the drinking behaviour of dairy cows. When located at the end of a common corridor, some subordinated animals stayed up to 48 h without accessing water [41].

Because paddocks should be designed to allow for prompt access to water and shade, VRG systems have been combined with other types of regenerative agricultural systems, such as silvopastoral. The integration of the silvopastoral system with VRG is fully compatible. The presence of silvopastoral nuclei within VRG paddocks improves the ambience and welfare of dairy cows [42], regulates the microclimate [42], sequesters C in soils [43], increases biomass [44], improves nutrient cycling [43], and provides habitat for pollinators and other species [45,46,47].

## 4. Voisin Rational Grazing Responds to Current Global Challenges

### 4.1. Climate Change

Cattle are thought to be a threat to climate change, mostly because of enteric methane (CH_4_) emissions. This reputation was mainly propounded as a consequence of GHG inventories from Life Cycle Assessments (LCA) based on guidelines of the Intergovernmental Panel on Climate Change (IPCC) Tier 1 [48,49]. These guidelines are essential tools to understand emissions, but they have some limitations. They presume that soil C is in equilibrium [50] and may underestimate grassland C sequestration potential and sink capacity [51,52], overestimating the global warming potential of CH_4_, a short-lived climate pollutant [53]. This might lead to false conclusions, such as feedlots having lower potential to climate change per kilogram of product than grass-fed systems [54,55,56,57,58]. LCA is the best internationally accepted approach to model and measure the potential impact of a given product but it may fail to achieve the actual positive impact of C sequestration on C balance of sustainable livestock systems, as is shown when soil organic carbon change is directly measured in soils.

Stanley et al. [59] used the same guidelines to consider C sequestration. When C sequestration was not sampled, they found that the adaptive multi-paddock grazing system (AMP) emitted 1.6 times more CO_2_ eq than grain-based systems. However, when soil C was systematically taken into consideration, net GHG fluxes resulted in an overall negative 6.65 CO_2_ eq kg^−1^ carcass weight (net C sequestration) for AMP and a positive 6.12 CO_2_ eq kg^−1^ carcass weight (net C emissions) for the feedlot system. These differences in C fluxes between systems indicate that well-managed pastures can offset emissions from the finishing phase of beef production. In fact, on-farm data showed that AMP had 13% and 9% more soil C and N, respectively, than conventional or set-stocking-managed grasslands [60] and that VRG grasslands stocked 25% more C than non-tillage fields for grain production [61].

Converting degraded agricultural areas to intensively managed grasslands has a high potential to mitigate climate change through fast increase in soil C. MacHmuller et al. [62] have determined a sequestration of 8.0 Mg C ha^−1^ yr^−1^ at peak accumulation near the sixth year after the conversion from cropland to intensively managed pastures. Putting this into perspective, only agroforestry performs better than this [63,64]. Other studies also found negative C balance for AMP with values for C sequestration of 3.53 and 3.59 Mg C ha^−1^ yr^−1^ [59,65]. However, soil is not a perpetual C sink [66]. It is part of the C cycle, and C storage is sensitive and reversible. C accumulation in a given site diminishes over time as levels of soil organic C content build up. Thus, caution should be applied in extrapolations and comparisons since estimations are always associated with their concomitant uncertainties [67].

Although some regenerative systems can maintain C sequestration for decades (e.g., [65]), others may arrive at an apparent plateau around 6 years following land use change [62]. The saturation of soil C depends on pedoclimatic, vegetation, and management characteristics. When this occurs, the area will become a source of C emissions owing to enteric CH_4_ emissions, a characteristic of livestock production. However, this will only happen when soil achieves a high organic matter content, so all the benefits from a sustainable system and healthy soil are available [62]. Under these conditions, VRG has its maximum potential to mitigate climate change when applied in highly degraded soil where C has been lost. The loss of 50% of the world’s agricultural land to soil degradation [68] affords an opportunity to sequester C from the atmosphere into the soil using adequate grazing management practices.

#### Reducing Emissions

Reduction of GHG emissions by the livestock sector mainly depends on reducing enteric CH_4_. Of the total livestock emissions, 40%, on average, comes from enteric fermentation, although manure management (N_2_O and CH_4_) is also relevant [69]. Secondarily, the other critical sources of emissions to the environment are from concentrate and synthetic fertilizers used to grow grains and pastures [60,70,71].

CH_4_ emissions from animal maintenance cannot be removed through management practices. However, it is possible to dilute animal maintenance through strategies that improve productivity and that may have strong efficiency in reducing GHG emissions on a per product basis [72]. As animal production increases, the proportion of energy and nutrients required for maintenance decreases, allowing for a greater proportion for milk or meat production [73]. However, one should always take into account environmental externalities and other production costs related to increased productivity. Animal breeding to increase animal productivity is key to reducing CH_4_ emissions but may also reduce animal resilience to environmental challenges in low-input farms or pasture farms [74]. Herd level management strategies to increase overall productivity (or reduce production inefficiency), e.g., culling unproductive heifers and administering vaccines, may also reduce livestock emissions [7].

Moreover, CH_4_ emissions can be lessened through management practices that provide diets of high nutritional value, to optimize animal productivity, leading to a reduction in enteric CH_4_ emissions. Diets with high fibrous carbohydrates slow enteric digestion and increase the CH_4_ emitted per unit digested when compared with feed with high concentration of starch and sucrose, such as grains, characterised by fast digestion [75], increasing rumen acidity, and thus reducing the emission of CH_4_ per animal unit. Grass-based diets cannot equal grain-based emissions based on these digestive conditions. However, by using grains as feed, one must acknowledge the share of deforestation responsibility for soy production in the Amazon [70,71], as well as in Brazilian Cerrado [76]. Neither can the spatial dependence of an indirect relationship between the expansion of croplands over pastures and further deforestation in neighbourhoods for new pasture areas be neglected [77]. Additionally, some studies point out that the hotspot for GHG emissions is the use of concentrate [78] and fertilizers in the production of grains [56]. Besides that, using grains ignores the importance of ruminants in converting food otherwise unsuitable for humans, such as cellulose, into high-quality protein [79]. Alternatively, VRG presents not only a high daily productivity per area, but it also maintains pasture in a vegetative state for a longer period, producing less fibrous forage [20] with the resultant lower potential for enteric CH_4_ emissions [80] than that in other grass-based systems.

When comparing grazing systems, DeRamus et al. [81] collected direct enteric CH_4_ emissions and demonstrated that MIG emitted 22% less CH_4_ annually when compared with continuous grazing systems. Effectively, recent in vitro results suggest that CH_4_ emissions can be minimised when herbage is at ORP (Figure 1). This is the time when plants have the highest protein and lowest fibre content associated with the highest herbage mass accumulation rate [80]. This shows that timing in grazing management is crucial to decrease CH_4_ emissions from grazing livestock. The end of the ascendant phase of the sigmoidal growth curve of plants is the ideal target to start grazing, which is also efficient in mitigating CH_4_ [7,80]. Intensification plays an important role on a global scale; however, owing to its effect on climate change, appealing to intensification of conventional livestock farming (or conventional agriculture in general) may cause equally important environmental impacts, such as acidification, eutrophication, and energy demand, and these effects are positively associated with the degree of intensification of production [82,83].

Other technologies can be combined and applied to grassland management. Feed additives have been used to reduce CH_4_ emissions; compared with stearic acid and soy oil, linseed oil can reduce enteric CH_4_ emissions from pasture-fed dairy cows [84]. Studies show that seaweed (*Asparagopsis* spp.) added in feed reduces from 80% up to 98% of CH_4_ emissions from livestock, depending on feed digestibility [85,86]. Other feed additives, such as 3-nitrooxypropanol (3-NOP) synthetic product [87] and other alternatives have been studied such as slurry chemical amendments [88] and protected urea [89,90,91] to reduce overall livestock GHG emissions. However, most are still being studied. They are also costly, and their uptake by farmers remains uncertain. The above options are useful as a palliative to minimise specific processes to reduce GHG emissions. However, we encourage more holistic options that address the underlying causes of global warming and move towards sustainable husbandry systems.

### 4.2. Ecosystem Services

#### 4.2.1. Maximising Carbon Sequestration and Storage in VRG

It is not enough to just reduce emissions; it is also necessary to maximise C stocks in soil. Suppose our goal in livestock production involves sequestration and storage of as much C from the atmosphere as possible. VRG management practices are precisely those recommended to do just that [64,92,93].

VRG grasslands are managed to minimise losses of C from soil through minimal soil disturbance before and during planting, maximising ground covering plants all year, and allowing adequate resting periods [94], as well as good design of corridors and tracks and judicious use of external inputs [95] such as fertilizers, supplemental feed, and the use of perennial species [96].

Moreover, VRG not only maximises the rate of C sequestration in biomass through ORP management, but it also includes other techniques that increase this rate, such as use of forage legumes, inclusion of especially deep-rooted plants to improve nutrient uptake, the increase in species diversity in pasture [97,98], introduction of fast-growing trees for shade and forage purposes, introduction of perennial species, and the use of inoculants [99]. Most effectively, integrating trees and pastures (silvopastoralism) makes it possible to neutralize CH_4_ emissions [92] by requiring a low number of trees to offset emissions from livestock production (17 to 44 trees ha^−1^; [63]).

Additionally, VRG maximises rates of transfer of C from biomass into soil [61]. In a recent study, Mosier et al. [60] reported a greater stabilization of soil C stocks in the AMP system when compared with conventionally grazed grasslands. Grazing high stocking densities associated with short grazing occupation returns much of the C ingested in forages back to the land in animal manure, basically pumping C back into the soil [10]. Systems could also use swales/terraces, avoid steep sites, use windbreaks to reduce wind erosion and de-compact compacted soil using deep-rooted plants [99]. Livestock management then becomes a tool for C sequestration.

#### 4.2.2. Soil Health and Biodiversity of Swards

In the VRG system, productivity and nutritive value of herbage produced, nutrient and hydrological cycles, as well as ecological services, all depend on the overall quality of soil, or health, and ultimately, therefore, the type of management practices used. Soil health is defined as the capacity of a living soil to function, sustain plant and animal productivity, maintain or enhance water and air quality, and promote plant and animal health [93]. To meet the requirements of this definition, VRG systems maximise aerial and below ground biomass production through the four principles previously mentioned as the main fluxes of C and nitrogen sources into the system.

Since living organisms within soil comprise a key pillar of VRG, VRG practices target improving the dynamics of microbes in soils. Science has only begun to understand the factors regulating soil microbiome and soil nutrient availability as a broader concept beyond that of a property characteristic of an external medium to which plants adapt [100]. Plant nutrition is a result of a functional, whole plant–soil–living organisms ecosystem where biodiversity underground affects several ecosystem functions [101]. Basically, the VRG approach focuses on nutrient management through microbial and plant-mediated processes [102,103,104,105]. Therefore, management practices are built on premises that rely on minimum soil disturbance and maximising C inputs. Processes in soil are undertaken by living roots and microorganisms sensitive to management. When living roots exudate liquid C, it feeds the microbiota that creates nutrient cycling and makes nutrients available to plants [103]. Such exudates create soil aggregates which open space for water infiltration and gas exchange, allowing microorganisms growth and further accelerating the process of organic matter accumulation. Soils with reduced disturbance have increased earthworm activity [106], further contributing to soil aggregation and increasing water entry into soil. Organic matter and living organisms in soil not only improve nutrient cycle dynamics, but also the water cycle since building aggregates helps water infiltration and retention. Accumulating soil C improves soil cation exchange capacity, water holding capacity and infiltration rates [8,62].

VRG also aims to minimise the utilization of external inputs, such as N fertilizer. Within this context, legume forage usage is promoted as it plays a key role in incorporating N into the system through N biological fixation. To maximise this effect, the ORP of legumes is frequently followed, aiming to favour the natural growth disadvantage that these species have in terms of photosynthesis rate, persistency, nutrients uptake, and growth rates when compared with grasses [107,108,109,110]. Effectively, recent research has shown that legumes can improve overall pasture multifunctionality, playing a key role in improving herbage production, N cycling, and herbage nutritional value when compared with monocultures [111].

Promotion of grassland ecosystem biodiversity is crucial to further enhance the system’s soil health. Under VRG systems, the use of permanent multispecies swards with perennial species of different families is paramount to maximise both the individual benefits as well as the synergistic benefits that the species can attain. When compared with monocultures, multispecies swards can: (a) increase overall herbage production, commonly showing increased biomass overyielding [98,111,112,113,114] and increased root biomass [115,116]; (b) better exploit the soil column and its available nutrients [115,116,117], improving overall nutrient utilization, reducing nutrient loses [118] and improving soil physical properties (e.g., density, structure and porosity), owing to complementarity in species radicular system characteristics [115,116,117,119]; (c) improve soil micro and mesofauna biodiversity [112,120]; and (d) have the potential to reduce soil erosion and reduce the presence of unsown species [113,114], as a consequence of higher spatial and temporal soil cover and protection [112,118].

The increase in overall system biomass productivity has been related to species seasonal growth asynchrony [117] and other complementarity scales [121]. Forbs play an important function in multispecies swards as well. For example, under temperate climates, some forb species (e.g., *Plantago lanceolata*, *Cichorium intybus*, *Achillea millefolium*, and *Taraxacum officinale*) have been included in the seed mix used in mixed swards due to their deep rooting system [117], high nutrient content, especially minerals that result in better animal nutrition [122,123], and anthelmintic properties [123,124]. On the other hand, in tropical climates, the presence of forbs increased resilience to drought, fire, and herbivory of pastoral ecosystems in a dynamic coexistence regime between grasses and forbs [125].

VRG typically uses multispecies swards, and the observation of ORP may result in higher root dry matter production (kg/ha) at 0–5 cm depth of the soil. The variable interval (ORP) averaged 7282 kg of DM/ha, while treatments of 21 and 42 days fixed cutting intervals averaged 6065 and 6404 kg of DM/ha, respectively [20]. Multispecies swards can also improve the hydrological cycle, reducing water runoff, increasing water infiltration, conferring potential tolerance to droughts [126,127], and increasing the variability of water supply, e.g., less frequent, but more intense, rainfall [128].

### 4.3. Food Quality

Human health can benefit from grass-fed dairy and beef products by their fatty acid composition. Individual fatty acids may have different roles in human health, and they can influence not only cardiovascular disease but also a range of other diseases, including metabolic diseases, such as type 2 diabetes, inflammatory diseases, and cancer [129]. High levels of saturated fatty acids (SFA) may increase the risk of cardiovascular diseases, coronary heart disease, and type 2 diabetes. On the other hand, some major monounsaturated fatty acids (MUFA, palmitoleic, and oleic acids) and polyunsaturated fatty acids (PUFA, linoleic acid) are shown to lower low-density lipoprotein (LDL) cholesterol concentrations associated with lower cardiovascular risk. Other PUFA omega-3 fatty acids are likely to prevent cardiovascular disease and a number of metabolic and inflammatory diseases. They may also have a role in the prevention and treatment of cancer [129].

Furthermore, increasing the proportion of grass in the diet resulted in a decrease in SFA, whereas MUFA, PUFA (mainly C18:3 n-3), and conjugated linoleic acid (CLA) increased linearly [130]. When fed grass, ruminants produce milk and meat with different profiles of secondary compounds, e.g., CLA and omega 3 [131,132], which have been linked to better human health [133,134], but see [135]. The change in milk FA composition from grass in the diet is also an opportunity to shift not only to a more nutritious source of food, but also to the consumption of nutraceutical diets.

Although grass-fed cattle under different grazing systems have been shown to produce a healthier milk and meat regarding FAs content, differences between organic and conventional pasture systems have not been found [136]. However, irrespective of conventional or organic, the stage at which the pasture is consumed affects the bioactive compounds of the herbage. Under the VRG system, comparing the content of secondary metabolites in the species *Avena strigosa* and *Lolium multiflorum* in three cutting intervals (38 and 54 days and a variable interval used by the farmer), Kuhnen et al. [137] found that the latter interval resulted in lower levels of carotenoids, flavonoids, and phenolics on pasture since the pasture had already passed the ORP which was close to the 38 d interval. Variations in bioactive compounds on pasture composition owing to its phenological stage has consequences on milk composition. A high correlation was found for values in milk and pasture samples in total phenolic content and ferric-reducing antioxidant power [138]. Pasture used at its ORP is richer in bioactive compounds, and this is likely to positively influence the quality of milk and meat and, therefore, human health.

### 4.4. Animal Productivity

When compared with other systems, VRG is considered to have the disadvantage of lower animal productivity. A study among small dairy farmers in the western part of Santa Catarina, for example, reports average milk yield of 10 kg/cow·day^−1^ on VRG farms [139], much lower than the semi-intensive systems in the same region (21–30 kg/cow·day^−1^; [140]). However, some VRG farms reported average daily milk production greater than 21 kg/cow·day^−1^ [140], which equals 6400 kg/lactation, far above the average of 2800 kg/cow·year^−1^ for the region [141] and even higher than the average of dairy production from Ireland (5438 L/cow·year^−1^ in 2018; [142]) and New Zealand (4296 L/cow·year^−1^ in 2019; [143]). The study from Balcão et al. [140] in western Santa Catarina also showed VRG presenting the highest milk yield response per kilo of concentrate offered (5.3 kg of milk/kg of concentrate vs. 3.7 kg of milk/kg of concentrate for continuous grazing and 4.0 kg of milk/kg of concentrate for semi-intensive).

Only a few studies have compared VRG with other systems on animal productivity for beef and sheep. A comparison between conventional grazing and VRG found a higher average daily gain (ADG) for the former (930 vs. 835 g/day), but a higher stocking rate (844 vs. 560 kg/ha) and forage production (95.5 vs. 60 kg/ha·day^−1^ of DM) for VRG and, hence, a 23% higher weight gain per area [144].

High animal yields on VRG farms can be obtained by applying the third and fourth principles of VGR (Table 1). On dairy farms, it is recommended that the lactating group of cows enter a fresh paddock after every milking, thus using two paddocks per day. Compared with lactating cows using depleted pasture, e.g., returning in the afternoon to the same paddock used in the morning, a significant difference in milk production was found, regardless of temperate (*Lolium multiflorum*, 28.6 vs. 26.2 kg/day) or tropical (*Pennisetum clandestinum*, 23.2 vs. 21.6) grasses [145]. The authors also found a higher pasture DM ingestion when using fresh paddocks, but a reduction in pasture DM ingestion with increased amounts of concentrate offered. No interaction between pasture state and concentrate level was found for intake, digestibility, or milk yield. For the harvesting of all available pasture, a second group of nonlactating cows and heifers should be moved to the paddocks immediately after the lactating cows leave it, providing tight grazing management.

A larger stock of C in pasture soil than in cropland is often questioned by the fact that one hectare of crop could produce more food than one hectare of pasture [146]. However, when comparing the potential of one hectare of pasture managed under VRG with cropland, Séo et al. [61] found a better result for VRG. They considered data collected in the field, plus requirements and feed composition from NRC Dairy tables, to estimate the potential for milk production in 1 ha of corn, followed by ryegrass vs. 1 ha of VRG pasture. It was estimated that VRG could produce 17,085 kg/ha·yr^−1^ of milk and that cropland could only produce 12,240 kg/ha·yr^−1^ of milk, given that all feed is destined to lactating cows.

These few studies show the need for more information on animal productivity in VRG and may be a critical knowledge gap in VRG. Anecdotally, farmers report high ADG when animals are first grazers. In southern Brazil, a group of 18-month-old heifers achieved an average ADG of 1.1 kg/day, and in Argentina, an average ADG of 0.9 kg/day was reported for 18-month-old Angus steers. In both cases, cattle were eating exclusively pasture. On the other hand, evidence on higher pasture productivity and quality in VRG was already obtained, as described above and in accordance with Voisin’s principles.

### 4.5. Farm Net Income

With more pasture becoming a component of feed, production costs decrease. An eight-year survey (2008–2015), with an average of 257 specialized dairy farms each year and 2055 surveys in total, was conducted in Ireland [147]. The authors found that increasing pasture use and length of season were significantly associated with an increase in net profit of USD 204.25 per ton of DM. On the other hand, when the proportion of purchased feed was increased by 10%, a reduction in net profit per ton of fat and protein of USD 244.39 was realized. Likewise, capital investment in machinery and buildings was negatively associated with net profit.

Farm profitability has been positively associated with the level of adhesiveness of VRG’s four principles and its refinements, such as water and shade or using two groups of animals [148,149]. This is in line with results from other intensive grazing systems where practices to improve pasture nutritional value indicated higher profitability. Thus, well-managed grazing systems can be more profitable than zero-grazing systems [150,151]. It seems that reduced feeding costs is one of the major drivers of VRG’s increased profitability [152]. Supporting this view is a study in the USA. When compared with zero-grazing, it was found that VRG systems yield a USD 64 greater net return per cow and that this was driven by lower feeding and labour costs [153]. Furthermore, grazing systems are likely to increase animal longevity, which is related to higher profitability, as was demonstrated in an Austrian study on organic farms [154].

Since the implementation of VRG can reduce feeding costs and external inputs such as chemical fertilizers, farms tend to be more financially resilient [155,156]. A simulation study, using cost and milk price data from Santa Catarina, Brazil, reported that farms of the same herd size (53 cows) using VRG and semi-confinement (see Balcão et al. [140] for definition of semi-confinement) systems producing 16 and 22 kg of milk/cow·d^−1^, respectively, would yield a yearly net income of USD 35,749.79 in the VRG system vs. USD 18,342.34 in the semi-confinement system [156]. Through sensitivity analysis, the same authors concluded that the VRG system is more resilient owing to lower production cost. This is in line with farmer’s perceptions since the use of ecological grazing management practices allows more resilience, independence, and success [157].

### 4.6. Environmental Externalities

External costs are relevant in livestock systems [158,159] and by definition, they are not paid directly by the producer, but paid by the society [160]. These environmental costs are multifactorial and difficult to assess. Some studies estimate that the annual cost of environmental externalities in U.K. agriculture between 1990 and 1996 was equivalent to GBP 208/ha for arable and permanent pastures [159]. In New Zealand, the estimated cost of environmental externalities of dairy activity exceeded NZD 11.6 billion [160]. It is necessary to include the external costs in net income calculations and this could tip the scales toward more ecological livestock systems.

On the other hand, VRG as other regenerative livestock grazing systems, may explore the possibility of profiting from ecosystem services. Carbon credits are already a viable option, but it mostly focuses on actions for maintenance, recovery, and improvement of vegetation cover in areas considered a priority for conservation. Credible and reliable measurement and monitoring platforms should be considered to report sequestered C for emissions trading [161], especially increases in soil C on a large scale through regenerative grassland management systems. Profitability is potentially the farmer’s main goal, and payment for C sequestration in livestock systems is more in line with farmers’ motivations [9]. Broader areas could be sustained by environmental services if greener practices could profit farmers by directing farmers’ focus on climate change mitigation management options.

### 4.7. Animal Welfare

High levels of animal welfare are reached if the animals are in good health, able to fulfil their behavioural needs, and experience positive emotions [162,163]. In VRG, grazing management is planned to offer high-quality nutrition from multispecies pastures, and paddocks are designed to have water and shade available at all times, minimising the effect of animal dominance over resources, i.e., resources are placed in ways that maximise the access of subordinated animals (e.g., [41]). Therefore, animals are able to fulfil their ethological needs of grazing and having outdoor access [164,165] forming social bonds [166,167] and grooming freely, a behaviour that is highly valuable for some ruminant species [168].

Tight grazing management exposes the lower parts of the plants and soil to solar radiation, and thus, time to return to the same grazed area, directly impacting parasite burden by reducing larvae survival and, hence, less infestation of animals [169]. This, combined with the opportunity to forage different plant species, may help animals to better cope with parasites since some plant toxins may reduce parasite proliferation [131,132,170]. Such variation in the environment, e.g., multiple forage species and terrain features, has been proposed to be a source of eutress (“good stress”) in grazing animals [171] since navigating the environment for diet selection, resting, and socially interacting may cognitively enrich animals’ lives. Animal agency has been proposed as a key element in the animal welfare concept [172]. It seems that many regenerative grazing systems, VRG included, provide the conditions needed for animals to achieve maximum welfare [173], as long as high standards of animal husbandry and veterinary care are provided.

## 5. Voisin Rational Grazing Challenges and Future Directions

Some challenges evidenced in VRG and by the farmers that have adopted this system are common for other types of regenerative grazing systems. VRG practitioners have experienced social constraints because of low acceptability and support [10,174] from: (1) livestock stakeholders that do not recognize VRG as a system that maintains high productivity along with the provision of other ecosystem services that last for mid/long term, since their expectation is a short-term productivity-oriented system; (2) the agro-industrial complex that relies upon the production and use of commercialized industrial feed products and agrochemicals, whereas regenerative grazing farmers pursue the reduction or elimination of all agro-feed and chemical products; and (3) peers and the local community of the farmers who have not implemented regenerative systems.

VRG requires a paradigm change in the grazing system, and such a shift is a big challenge for ordinary farmers [11,175]. The successful implementation and maintenance of regenerative grazing systems, such as VRG, depends on a farmer’s conviction in the system, a conviction that arises from a deep understanding of the ecological processes involved in the system, along with a set of skills related to monitoring and planning livestock grazing [10]. The success of these ecological production systems is usually facilitated through the creation of a sense of community [157]. Indeed, several networks have been created to support farmers, extensionists, and researchers to foster more sustainable grazing systems (e.g., Soil Health Institute https://soilhealthinstitute.org, Savory Institute https://savory.global, Agricultura Regenerativa Ibérica https://www.agriculturaregenerativa.es, Pasture Fed Livestock Association https://www.pastureforlife.org, and Nucleo de PRV/UFSC https://nucleoprv.paginas.ufsc.br/?lang=en). However, these initiatives should be accompanied by complementary regional level policies as they are key in helping to sustain agroecological initiatives [176].

Farmers perceive VRG as a high-capital and time-investment system based on infrastructure costs and essential management capacitation. Despite global initiatives to promote agroecology (e.g., FAO), only minimal financial support has been forthcoming for new agroecology-oriented research, education, project implementation, and maintenance since such techniques are not in line with common industrial and Green Revolution agricultural practices [177,178]. This has resulted in low recognition of the benefits that agroecological-based systems can provide society from government and agricultural stakeholders alike [170,176].

Although many commercial VRG projects can be found in vastly different pedoclimatic conditions, from cold temperate to tropical regions, the research related to VRG outcomes and difficulties faced by farmers have not experienced similar progress. No systematic record of information has arisen to support the results obtained. Additionally, very few experimental research stations exist. To overcome these challenges, multicentric studies should be developed to validate the empirical evidence generated on commercial farms.

VRG results in increased herbage productivity immediately or a few years after implementation. However, some farmers experience two to five years of herbage productivity stagnation, named “years of misery” by Voisin [17], which generally occurs during the soil recovery process from previous management practices (e.g., soil compaction) to build up stable soil microbiological activity [12]. Because of this prolonged period, many adopters of VRG may give up this system and transition back to conventional farming. Pinheiro Machado [12] suggested strategies to minimise this period of low pasture production, including the use of organic fertilizer (e.g., compost, poultry litter, and swine manure), combined with the four principles of VRG.

Another management problem experienced by VRG farmers begins with the high herbage growing rates observed commonly during spring/summer/wet seasons that is suddenly reduced during fall/winter/dry seasons. When this change is not followed by a reduction in grazing intensity to compensate for the expected seasonal fluctuation, a continued shortening of the ORP, known as “untoward acceleration” [17], occurs. This untoward acceleration goes against VRG’s first principle, hampering the overall accumulation of plant reserves and compromising future herbage recovery and productivity. To overcome this challenge, the farmer can apply several techniques for forage conservation, producing silage and hay with the surplus herbage produced during the high-growth season

To help with challenges in VRG, technology can be applied. The use of satellite images, drones, and other sensors to monitor herbage production and phenological stage and aid in identifying OPR have been recently studied and have yielded interesting results (e.g., [179,180,181]). Moreover, animal behaviour monitoring used in conventional livestock agriculture (e.g., [182]) can be used to improve animal productivity and health. Additionally, the use of new technologies may drive young farmers to continue in the field, helping to avoid more rural exodus.

Finally, a structured market based on reliable measurements and monitoring platforms is urgently needed to report ecosystem services in order to allow for payments to farmers. Rewarding those who apply greener practices that benefit society as a whole may incentivise more people to shift towards regenerative practices.

## 6. Conclusions

VRG is a viable option for livestock farming, given current global challenges. VRG is based on four simple principles arising from plant and animal physiological requirements. As a consequence, it is able to sustain regular productivity, and it is ultimately profitable. VRG also delivers important ecosystem services, increasing soil C uptake, improving soil health, boosting water retention, protecting water quality, and fostering biodiversity. However, VRG is no panacea and comes with challenges that require patience to overcome, as well as a set of actions at the farm, local, and global levels.

## Figures and Tables

**Figure 1 animals-11-03494-f001:**
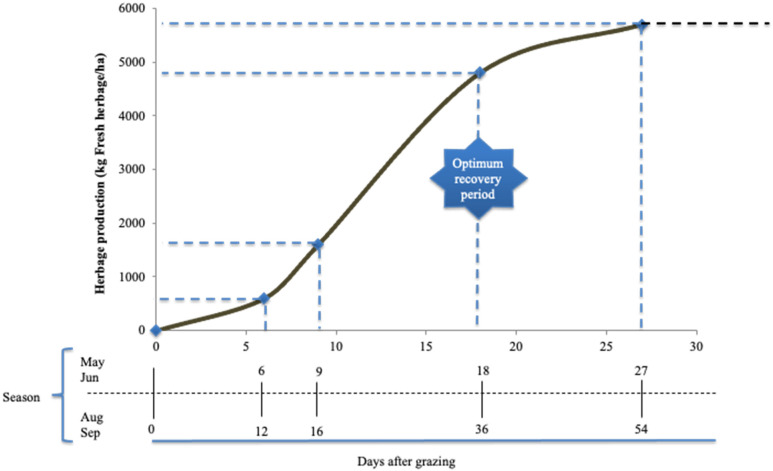
Plant growth curve (adapted from Voisin, [17]).

**Figure 2 animals-11-03494-f002:**
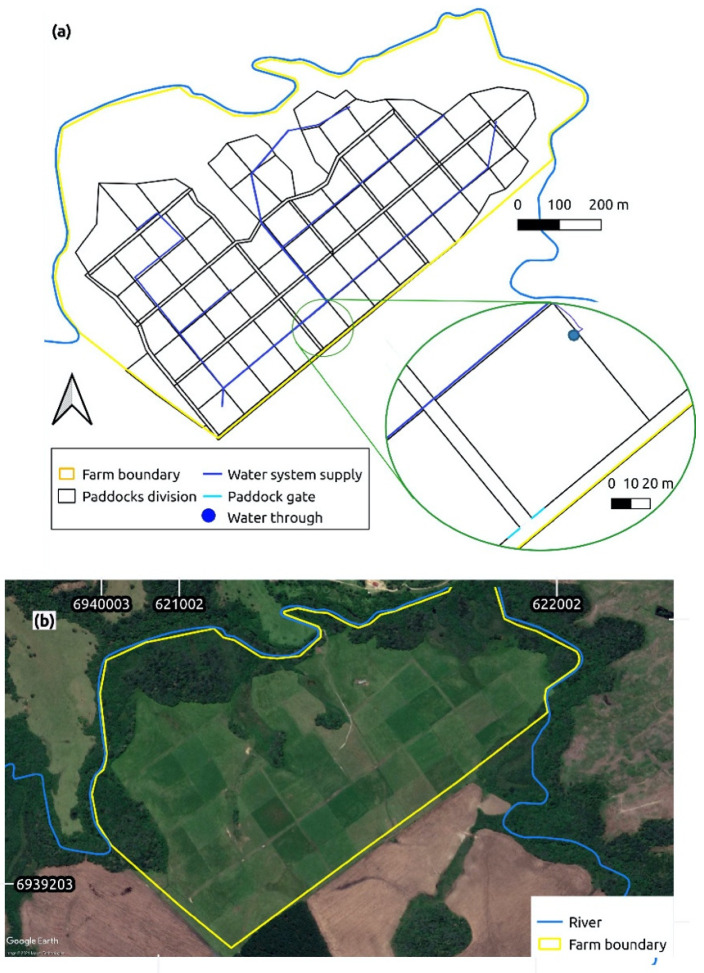
Eight-year-old commercial beef Voisin Rational Grazing (VRG) farm in Bom Retiro, Santa Catarina, Brazil. The total pasture area is 45 ha. (**a**) Blueprint of paddocks, alleyways (corridors) and hydraulic system. The total number of paddocks is 68. In VRG, water goes to the animals [12]; (**b**) satellite footage (Map Source—QGIS 3.16. (November 2018), Google Earth, Maxar Technologies, of the farm (accessed on 16 August 2021). The non-sequential use of paddocks creates what is called a “chess pattern” [12].

**Table 1 animals-11-03494-t001:** The four “laws” (principles) of Voisin Rational Grazing.

Principle (Law)	Goal(s)	Description/Management
(1) Recovery period	Maximum pasture productivity and restoration of reserves	Observe the correct ORP ^1^ in order to allow maximum herbage productivity, high forage quality and reserve storage for following regrowth. The period of rest of the grass between two successive cuts will be variable according to the plant species, season of the year, climatic conditions, soil potential, and other environmental factors.
(2) Occupation	Avoid cutting early regrowth, promote soil biocenosis and grazing efficiency	Observe high stocking densities for a short period of time to prevent grazing of plants in early regrowth and to deposit large amounts of manure. Apart from exceptional situations, occupation time should not exceed 3 days, and ideally it would be 12 h for dairy or 1 day for beef.
(3) Maximum performance	Increase animal productivity	Allow animals to graze pastures of nutritive value that match their nutritional needs. Split the herd according to the nutritional needs of the animals into 2 or 3 groups, moving firsts, seconds, and thirds in sequence in all paddocks.
(4) Regular performance	Ensure regularity in animal productivity	Observe short periods of occupation per group to provide regular pasture allowance according to the animals needs and constant nutrient intake.

^1^ OPR: optimum recovery period.

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
