# Peer review of "Voisin Rational Grazing as a Sustainable Alternative for Livestock Production"

_animals, 2021, doi:10.3390/ani11123494_

Round 1

Reviewer 1 Report

Comments.

Voisin Rational Grazing as a sustainable alternative for livestock production

The paper is set up correctly — The referee give some suggestions to improve the presentation of the manuscript and minor remarks, which should not prove too difficult for the authors to modify. Please consider to include recent publications to the review. Some of them are very old and new information is available.

Detailed comments

Simple summary,

Add at the end of the paragraph the “constrains or limitation” of VRG. The same observation for the abstract section.

Introduction

Lines 76-85. Briefly explain the grazing managements that the authors mention. Better explain the differences among them and the particularities of the VRG.  Explain the advantages (if this is true) of VRG over the other systems.

Lines 95-97

Name the four principles of VRG

Lines 113

There is an evidence of that? please mention the source.

Lines 161-162

“the ORP of a particular species can be targeted to allow it to increase its presence in the paddock [11].”

This is true, but better explain how this could be possible.

Lines 162-165

“Moreover, targeting the ORP of the most productive and/or best quality plant species may promote ideal conditions to maximize overall annual herbage production and quality [11], in turn hampering the survival of undesirable species or weeds [26–28].“

This is the ideal situation, but may happen the opposite. Please elaborate again this paragraph and better explain the factors needed to achieve this effect.  

Lines 181-191

Rephrase the paragraph and better explain the sequential grazing management. Give an example to the reader to follow your idea.

Lines 206-212

Please indicate how to regulate the stoking rate between the leader and the follower animal groups. i.e. How the give plasticity to the management and to macht animal needs and plant production.

Lines 242-243

Please indicate the likely positive and negative effects on sward heterogeneity. Here, the authors indicate as advantage the loose of the number of species within the swards. This is not always true. Patchs might also have positive effects for biodiversity of insects by creating micro niches and better nutrient utilization. Please consider to include these aspects in this section.

Lines 243-249

Apart from aggressiveness and labor, which other disadvantages the author can mention about implementing VRG?

Lines 408

Split this section in two. First Soil health then biodiversity of swards

Lines 435-441

Explain how legumes are in disadvantage versus grasses

Lines 550-579

Unify the currency to have better idea of the comparisons among countries

Include a subsection of constrains and limitation of VRG

This is important to contrast it against the benefits of the grazing model

Author Response

Thank you very much for reviewing our manuscript. We found your contribution very valuable, and we tried to accomplish with all suggestions, or justify our point of view.

R1:

Comments and Suggestions for Authors

Comments.

Voisin Rational Grazing as a sustainable alternative for livestock production

The paper is set up correctly — The referee give some suggestions to improve the presentation of the manuscript and minor remarks, which should not prove too difficult for the authors to modify. Please consider to include recent publications to the review. Some of them are very old and new information is available.

AU: We thank you for your comments. We have reviewed the cited literature and updated it where appropriate.

Detailed comments

Simple summary,

Add at the end of the paragraph the “constrains or limitation” of VRG. The same observation for the abstract section.

AU: Added. Simple summary lines 28-29, now reads as follows: “Here, we provide a comprehensive overview of VRG, along with its benefits and constraints.”

Abstract: lines 42-47, now reads: "It can be said that VRG practitioners are part of the initiatives that are rethinking modern livestock agriculture. Its main challenges, however, arise from social constraints. More specifically, local incentives and initiatives that encourage farmers to take an interest in the ecological processes involved in livestock farming are still lacking. Little research has been done to validate the empirical evidence of VRG benefits on animal performance or to overcome VRG limitations.

Introduction

Lines 76-85. Briefly explain the grazing managements that the authors mention. Better explain the differences among them and the particularities of the VRG.  Explain the advantages (if this is true) of VRG over the other systems.

AU: We acknowledge the comment of the Reviewer and understand his/her request. However, mentioning a brief description of the different grazing managements may result in a biased and very imprecise characterization of the peculiarities of each of those systems. Thus, we prefer to mention the references where the reader can find a precise and more detailed description of each of the systems. Moreover, we do not aim to compare VRG grassland management or outcomes with any other system, but merely to discuss the potential benefits that VRG can promote, as stated in the objectives (Lines 101-104).

Lines 95-97 Name the four principles of VRG

AU: Named. Lines 97-99, now reads: “VRG focuses on four basic principles (recovery period, occupation time, maximum performance and regular performance principles; see section 2) that account for forage growth and management, as well as animal requirements.”

Lines 113

There is an evidence of that? please mention the source.

AU: We have added the reference for this. Lines 116 - 118 now read as follows: "Voisin’s work was first implemented in South America in 1964 by the agronomist Nilo Ferreira Romero on his farm called Conquista located in Bagé, southern Brazil [12,20,21]."

Lines 144-145:

AU: We also included: which has been related to the regrowth moment when light interception reaches 95% [23].

Lines 161-162

“the ORP of a particular species can be targeted to allow it to increase its presence in the paddock [11].”

This is true, but better explain how this could be possible.

AU: Please, see next comment.

Lines 162-165

“Moreover, targeting the ORP of the most productive and/or best quality plant species may promote ideal conditions to maximize overall annual herbage production and quality [11], in turn hampering the survival of undesirable species or weeds [26–28].“

This is the ideal situation, but may happen the opposite. Please elaborate again this paragraph and better explain the factors needed to achieve this effect.

AU: We acknowledge the reviewers comments and therefore, the following phrase has been added to add clarification to the mentioned processes. Lines 167-185 now reads as follows: Moreover, targeting the ORP of the most productive and/or best quality plant species may promote ideal conditions to maximise overall annual herbage production and nutritive value [11], in turn hampering the survival of undesirable species or weeds [26–28]. Such effects are related to the fact that when the plant is cut at its ORP, there is the best combination of accumulated reserves and the lowest fiber content in the plant tissue [31]. Thus, when cut at this point, it will have a faster and more vigorous regrowth than other plants that have not attained their ORP and will have fewer reserves to promote a vigorous regrowth. More mature plants that have passed their ORP, will have already redirected some of the accumulated reserves to the seed formation, and will be less palatable to animals due to a higher fiber content. As a consequence, the grazed plant will have a higher senescent residue, with greater respiration and lower photosynthetic rate during the regrowth [32], reducing their competitiveness compared to plants cut at their ORP. Pratensis plants, as Voisin (1957) called it, or plants growing in meadows that co-evolved with ruminants, have high tolerance to grazing. Compared to 60 days cutting interval, frequent defoliation reduced root and shoot biomass in species with high tolerance to grazing, but not in species with low grazing tolerance [33]. Thus, it is expected that in a multispecies pasture cut at its ORP, this characteristic would favour the presence of high tolerance grazing species and reduce the participation of non-grazing species.

Lines 181-191

Rephrase the paragraph and better explain the sequential grazing management. Give an example to the reader to follow your idea.

AU: Paragraph rephrased. Thank you for your comment. The paragraph now reads as follows (L205-212): To achieve maximum herd performance, animals of higher nutritional demand should be allowed the herbage of greatest quality. Herbage nutritional value is greatest at the top fraction of the canopy and lowest at the lower fractions [31,32]. Thus, to follow this rationale one may separate groups of animals by their nutritional requirements. For example, lactating animals (higher nutritional demand) may enter a fresh paddock while non-lactating animals (lower nutritional demand) may enter that same paddock shortly after the lactating herd has left to a new fresh paddock. This management is deemed first and second grazing groups [37].

Lines 206-212

Please indicate how to regulate the stoking rate between the leader and the follower animal groups. i.e. How the give plasticity to the management and to macht animal needs and plant production.

AU: We tried to clarify the issue. Please now read the paragraph as following (227-239): The dynamic and complete observance of VRG principles is key to attaining maximum system production efficiency, including positive responses in the quality of food produced [16]. The application of the four principles must be dynamic, dialectic, and constantly evaluated, but without fixed rules, fixed times, or fixed stocking rates. Thus, this requires good planning. It is a dynamic management process of the soil-plant-animal complex with holistic evaluation throughout the pastoral ecosystems. The key aspect to achieve such management is time. ORP never has the same length, therefore the sequence of paddocks´ use is not repeated in consecutive grazing seasons. Likewise, occupation time varies with pasture productivity over the season. In the first-second group dynamic, the first group will leave to a fresh paddock when all pasture of the second group´s paddock is consumed. Therefore, the second group defines the moment of paddock change for both groups. These management principles (Table 1) are oriented toward satisfying both herbage and animal requirements [16].

Lines 242-243

Please indicate the likely positive and negative effects on sward heterogeneity. Here, the authors indicate as advantage the loose of the number of species within the swards. This is not always true. Patchs might also have positive effects for biodiversity of insects by creating micro niches and better nutrient utilization. Please consider to include these aspects in this section.

AU: Thank you very much for your comment. In fact, we might have used the wrong words to express this concept. We didn’t mean losing the number of species, but cutting the patch in a more homogenous way, without leaving behind non consumed plants. In fact, the number of species in VRG pastures is fairly high. Therefore, we rephrased it as follows (269-280): “Under VRG, the animals are less selective in their grazing behaviour, becoming more voracious. This means that animals graze almost all species available, leaving few unconsumed species in the sward [11,14,25,27]. As plants are repeatedly cut in this way, there is a tendency to reduce the presence of non-grazing species, although not decreasing diversity. Azevedo et al. [40] report 81 plant species from 23 different families per square meter in a VRG in Bom Retiro, SC, Brazil. If the high animal density leads to a more voracious ingestive behaviour, applying excessive stocking densitymay affect overall animal behaviour and welfare. For example, comparing two high stocking densities (200 cows/ha and 500 cows/ha), cows in the lower stocking density group performed more grazing and had less aggressive behaviour than cows in the higher stocking density [41]. Furthermore, the use of a very high stocking density may require three or more paddock changes per day, increasing labour.

Lines 243-249

Apart from aggressiveness and labor, which other disadvantages the author can mention about implementing VRG?

AU: Aggressiveness and more labour are not “disadvantages” of VRG. Constraints of VRG are discussed in the paper in lines 656-722. In former lines 243-249, now 270-274, we were comparing two stocking densities in VRG, and found that very high instantaneous stocking rate may lead to more aggression among animals and require a higher number of paddock changes per day, thus potentially increasing labour. For a better understanding, we rephrased the sentence, as shown above.

Lines 408

Split this section in two. First Soil health then biodiversity of swards

AU: Thank you for your suggestion. The previous versions of our manuscript had this section split. However, we found that some of the passages were becoming repetitive. This arises because most of our discussion related to the biodiversity of swards is related to its impact on soil. To avoid repetition, we decided to keep this as one section, but we explicitly changed the section title to "Soil health and biodiversity of swards". (line 441)

Lines 435-441

Explain how legumes are in disadvantage versus grasses

AU: We have amended the phrase as suggested and also have included some references to broaden the level of our assertion (lines 472-475): "To maximise this effect, the ORP of legumes is frequently followed, aiming to favor the natural growth disadvantage that these species have in terms of photosynthesis rate, persistency, nutrients uptake and growth rates when compared to grasses; [115-118]"

Lines 550-579

Unify the currency to have better idea of the comparisons among countries

Include a subsection of constrains and limitation of VRG

This is important to contrast it against the benefits of the grazing model.

AU: Thank you for the suggestion. We have unified the currencies to USD using the exchange rates at the time of the publications. All changed in section 4.5/

Reviewer 2 Report

The manuscript is well organized with clear objectives,I think the authors did great effort to collect the information. For me this manuscript is suitable for publication.

Author Response

Thank you very much for reviewing our manuscript.

Reviewer 3 Report

Dear authors

I found the paper well written and organized, clear and pleasant to read. The topic if of importance especially in Latin America where frequently producers are questioning regarding this proposed management strategy. The manuscript presents the idea of the “Voisin rotational stocking” (explanation below regarding terminology) where four general rules (referred as “laws”) serve as parameter to base management decision.  Those rules are related to pasture growth and animal intake, although don’t offer specific parameters on what a management based on the principles proposed would look like. Furthermore, to the best of my knowledge, there is no scientific support to some of the assumptions on the rules, especially rule #4 (I will present some more comments further). The document covers a lot of ground on the benefits of well managed grazing systems (in a general sense); multiple of the references used would be similar to any other proposed rotational stocking management strategy, although I missed some of the core literature that could strengthen the discussion. Nevertheless, I don’t agree with the use of that for supporting one specific management strategy, as it would be common for any “well managed” pasture (using veery vague terms here).

The document sometimes lacks the extra step to convince readers that indeed the proposed VRG is what is making the difference. I believe some of this is due to the lack of scientific work on this management, specifically. However, multiple other works could be used and extrapolated. I will present more comments on this later. When seldom presenting research values, most of the contrast done is either with feedlot or semi-confined systems, or rotational stocking. In terms of stocking management, the actual stocking management strategy is second to concepts of carrying capacity and stocking rate. In other words, if forage allowance is not sufficient, simply rotating animals will not solve the hunger issue (See Sollenberger and Vanzant, Crop Sci 2011 51:420-432, for example).  At one point it is mentioned that “maximum performance of pasture and animal performance is obtained with the dynamic and complete …” (l. 222-223); first, no reference supporting such assumption; second, dynamic and complete seems to be conflicting here, because in my interpretation I have to obey to all the rules (laws) otherwise it will not work; that does not sound too dynamic. Similar statements are presented throughout the text without solid references or contrasts to proof, and they seem anecdotal.

Both sections 46 and 4.7 seem a little far reach for me (especially 4.6) as it does not provide why the proposed management would be different than other strategies regarding those responses. Although well written and interesting, they do not add to the main goal of the document. Same comment goes for section 4.3. On section 4.1, especially 4.1.1, I failed to identify any of the references mentioned which have indeed tested the VRG on any of the parameters discussed. Nevertheless, there is a good body of literature supporting some of the claims in other management strategies. A potential reference is Zubieta et al 2021 (Sci Total Environ754  142029), who recently presented an overview of the topic.

Regarding number of paddocks, in review dne by Sollenberger et al 2012 (in Conservation Outcomes From Pastureland and Hayland…; NRCS), half of the studies reported advantages from pasture quantity with increased number of paddocks while half had no advantage. The infrastructure cost, however, is much greater as we increase paddock number. I appreciate the authors identifying the lack of research on the topic and highlighting future needs on section 5. That is very professional and important for the continuation of the research on that topic. On the other hand, I notice a lack of some of the current American literature recently published on comparison of Management Intensive Grazing (or mob-grazing). Both on rangelands (e.g. Augustine et al 2020 Range Ecol Manage 73(6)796-810) and on grasslands (e.g. recent work done by Redden and by Gompert in Nebraska; and by Tracy in Virginia), the multitude of claimed benefits from adopting such management strategy compared to other rotational stocking were not supported – especially related to animal and pasture productivity. From the works in Nebraska on a 5-year study, daily gain of myearling steers averaged 1.49 lb/day for the 4-paddock system and 0.39 lb/day for the m120-paddock system and forage production was not different between the two treatments. Similar results were found in Virginina which also reported no benefit on soil C.

Although based on a limited number of studies conducted primarily in rangeland, there is no current consensus in the literature that mob stocking affects SOC accumulation rate differently than rotational stocking with fewer paddocks. If compared with continuously stocked pastures that are overstocked, advantages of mob stocking may be found (Chaplot et al., 2016), but these differences may be due to differences in stocking rate and herbage allowance rather than stocking method. (From Sollenberger et al 2019 Crop Sci 59:441-459)

In all, I believe there is a bias on the text and lack of results showing other strategies or the comparison between VRG to other strategies.

General comments:

Laws – according to a few definitions easily found online on the main streams of science, the use of the word “law” refers to 1)A descriptive generalization about how some aspect of the natural world behaves under stated circumstances; or 2) the description of an observed phenomenon. It doesn't explain why the phenomenon exists or what causes it; or even 3) statements, based on repeated experiments or observations, that describe or predict a range of natural phenomena. While theory refers to a well-substantiated explanation of some aspect of the natural world that can incorporate facts, laws, inferences, and tested hypotheses. I don’t think the use of the word “law” as proposed by the authors is a good fit. Set of rules, guidelines or parameters would be more approprate

Terminology – I strongly suggest following guidelines presented by Allen et al 2011 – Grass and Forage Sci 66, 2–28. They proposed referring to rotational “Stocking”  rather than grazing; also instantaneous stocking rate should be referred to “stocking density”. Quality (l 164) should be referred to as nutritive value, as “quality” itself includes intake components and is evaluated as performance. On l. 235 – leader and follower grazing groups would be first and second grazers

Specific comments on text:

  1. 76 km2 – the “2” should be upper script
  2. 108 rotational “stocking” method
  3. 295 – “-1” should be upper script
  4. 399 - “-1” should be upper script
  5. 428 – “open the pore spaces” actually aggregates create por spaces, not open them

Section 4.4 units are being represented with “/” instead of “-1” – check notation guidelines from journal

On the references, please check the following for completion, formatting and other issues (this was not a thorough review – just a few that I caught)

  • 6 – capitalized title
  • 8 – missing where it was published
  • 19 – idem
  • 23 – capitalized title
  • 136 – “no prelo” needs to be changed to “in press” or equivalent
  • 139 – symbols appear instead of letters in the names

Author Response

AU: Thank you for your comments and suggestions. We have extensively revised the manuscript to accommodate/address the suggestions we found pertinent. One should take into consideration that this manuscript is about VRG, and its aim is to describe VRG as one alternative to conventional livestock farming. Below we themed your major concerns in topics and provide a rebuttal to each individually, highlighting the changes made in the original manuscript when applicable.

Aim of this manuscript:

AU: The aim of this manuscript is to describe VRG as one of the alternatives to conventional livestock farming. We acknowledge that other well managed grazing systems may also positively respond to global challenges. This has been highlighted in the Introduction, lines 105-109: "In the first section, we highlight the principles that guide VRG, the refinements proposed by several researchers to improve its productivity and identify other grazing systems that are similar to VRG. Secondly, we demonstrate how VRG responds to current global challenges in livestock management. Thirdly, we discuss the barriers against the adoption of VRG. We finish with future directions and limitations."

VRG principles, scientific soundness and interpretation:

AU: We have amended section 2 to add clarity regarding the application of the four principles. In table 1 there are examples on how each principle may be achieved. New passages include (lines 227-239): "The dynamic and complete observance of VRG principles is key to attaining maximum system production efficiency, including positive responses in the quality of food produced [16]. The application of the four principles must be dynamic, dialectic, and constantly evaluated, but without fixed rules, fixed times, or fixed stocking densities. However, this requires good planning. It is a dynamic management process of the soil-plant-animal complex with holistic evaluation throughout the pastoral ecosystems. The key aspect to achieve such management is time. ORP never has the same length, therefore the sequence of paddocks´ use is not repeated in consecutive grazing seasons. Likewise, occupation time varies with pasture productivity over the season. In the first-second group dynamic, the first group will leave to a fresh paddock when all pasture of the second group´s paddock is consumed. Therefore, the second group defines the moment of paddock change for both groups. These management principles (Table 1) are oriented toward satisfying both herbage and animal requirements [18]."

The word "law" was the way it was first described by Voisin himself, and we used the word quoting Voisin. Anyways, for better clarification, we also used "principles" instead. 

Lack of specific scientific evidence for VRG systems:

AU: We acknowledge the reviewer's comment in this aspect. And this is a good reason to publish this review. During the last 50 years, VRG was repeatedly rejected by the “scientific community”, specially those in the Forage area. One of the possible consequences of this review is to arouse interest by researchers for the VRG, possibly resulting in more research to clarify many aspects of the system. As a matter of fact, this is one of the future challenges we pointed out at the end of the manuscript, in item “5.5. Voisin Rational Grazing challenges and future directions”.

We have highlighted the limited number of scientific works supporting some of the aspects discussed in the manuscript in lines 687 to 692. We identified AMP and MIG management as being similar to VRG (see Lines 242 to 248), thus we have quoted research that has been used with these two systems, as well as research done with VRG, throughout the manuscript to support the results and ideas mentioned in the text. To aid clarity, we have amended in L245 the following phrase "These management methods follow principles very similar to the four previously described for VRG, thus we will use the research outputs from studies undertaken using these systems, along with studies done with VRG, to support the rationale and mechanism underlying VRG responses". Regarding the particular concerns in section 4.1, despite using similar approaches quoting work undertaken with AMP and MIG, some work has indeed been undertaken using VRG (e.g. Pereira et al., 2020, Seo et al., 2017).

Far reach sections (4.3; 4.6 and 4.7):

AU: Thank you for pointing this out. Again, our manuscript is about VRG. We agree that others well managed grazing systems may deliver similar results as described in the manuscript, and this has been highlighted in all these sections:

Section 4.3 (lines 526-531): "Although grass-fed cattle under different grazing systems have been shown to produce a healthier milk and meat regarding FAs content, differences between organic and conventional in pasture systems have not been found [144]. However, irrespective of conventional or organic, the stage at which the pasture is consumed affects the bioactive compounds of the herbage. Under VRG system, comparing the content of secondary metabolites in the species Avena strigosa and Lolium multiflorum in three cutting intervals...";

Section 4.6 (line 623): "VRG as other regenerative grazing systems may explore the possibility of profiting from ecosystem services."

Section 4.7 (lines 654-656): "It seems that many regenerative grazing systems, VRG included, provide the conditions needed for animals to achieve maximum welfare [181], as long as high standards of animal husbandry and veterinary care are provided.”

We feel that as part of a narrative review it is important to keep these sections in the manuscript as they contribute to the general discussion about livestock farming.

General comments:

Laws – according to a few definitions easily found online on the main streams of science, the use of the word “law” refers to 1)A descriptive generalization about how some aspect of the natural world behaves under stated circumstances; or 2) the description of an observed phenomenon. It doesn't explain why the phenomenon exists or what causes it; or even 3) statements, based on repeated experiments or observations, that describe or predict a range of natural phenomena. While theory refers to a well-substantiated explanation of some aspect of the natural world that can incorporate facts, laws, inferences, and tested hypotheses. I don’t think the use of the word “law” as proposed by the authors is a good fit. Set of rules, guidelines or parameters would be more approprate

AU: Thank you for the suggestion. As already mentioned, this was the way it was first described by Voisin himself. That is why the word law appears quoted in the manuscript. However, throughout the text we used the word "principles" to avoid any misunderstanding.

Terminology – I strongly suggest following guidelines presented by Allen et al 2011 – Grass and Forage Sci 66, 2–28. They proposed referring to rotational “Stocking”  rather than grazing; also instantaneous stocking rate should be referred to “stocking density”.

AU: Thank you. We substituted the terminology according to the suggestions that fit our narrative and following the guidelines presented by Allen et al 2011, whenever possible (e.g. stocking density and first and second grazers)

We kindly disagree with the suggestion to use the terminology from Allen et al. (2011) to name VRG. Voisin Rational Grazing is a name and a concept, defined in the text “... as a rational method for managing the soil-plant-animal complex through direct grazing and well-planned pasture rotation. (L112-113)” Therefore, we decided to leave the name as is, once we do not agree that the terms used by “The Forage and Grazing Terminology Committee” can fully capture the meaning of VRG.

Quality (l 164) should be referred to as nutritive value, as “quality” itself includes intake components and is evaluated as performance.

AU: We realize the difference and agree that nutritional value is more precise in the referred situation. We changed the word throughout the manuscript.

On l. 235 – leader and follower grazing groups would be first and second grazers

AU: Thank you. Yes, changed.

Specific comments on text:

Line: 76: km2 – the “2” should be upper script

Line: 108: rotational “stocking” method

Line: 295: “-1” should be upper script

Line: 399: “-1” should be upper script

AU: Thank you. All these minor comments have been corrected.

Line: 428: “open the pore spaces” actually aggregates create por spaces, not open them.

AU: Yes, thank you. L460-464 now reads: “When living roots exudate liquid C, it feeds the microbiota that creates nutrient cycling and makes nutrients available to plants [111]. Such exudates create soil aggregates which open space for water infiltration and gas exchange, allowing microorganisms growth and further accelerating the process of organic matter accumulation.”

Section 4.4 units are being represented with “/” instead of “-1” – check notation guidelines from journal

On the references, please check the following for completion, formatting and other issues (this was not a thorough review – just a few that I caught)

 Line 6 – capitalized title

Line 8 – missing where it was published

Line 19 – idem

Line  23 – capitalized title

Line 136 – “no prelo” needs to be changed to “in press” or equivalent

Line 139 – symbols appear instead of letters in the names.

AU: Thank you very much for the remarks. Yes, we reviewed the manuscript regarding these details.